# *Arabidopsis* histone H3 lysine 9 methyltransferases KYP/SUVH5/6 are involved in leaf development by interacting with AS1-AS2 to repress *KNAT1* and *KNAT2*

Fu-Yu Hung [1,2,5], Yun-Ru Feng[1,5], Kuan-Ting Hsin[3], Yuan-Hsin Shih[1], Chung-Han Chang[1], Wenjian Zhong[1], You-Cheng Lai[1], Yingchao Xu[4], Songguang Yang[4], Keiko Sugimoto [2], Yi-Sheng Cheng [1,3] & Keqiang Wu [1✉]

The *Arabidopsis* H3K9 methyltransferases KRYPTONITE/SUPPRESSOR OF VARIEGATION 3–9 HOMOLOG 4 (KYP/SUVH4), SUVH5 and SUVH6 are redundantly involved in silencing of transposable elements (TEs). Our recent study indicated that KYP/SUVH5/6 can directly interact with the histone deacetylase HDA6 to synergistically regulate TE expression. However, the function of KYP/SUVH5/6 in plant development is still unclear. The transcriptional factors ASYMMETRIC LEAVES1 (AS1) and AS2 form a transcription complex, which is involved in leaf development by repressing the homeobox genes *KNOTTED-LIKE FROM ARABIDOPSIS THALIANA 1* (*KNAT1*) and *KNAT2*. In this study, we found that KYP and SUVH5/6 directly interact with AS1-AS2 to repress *KNAT1* and *KNAT2* by altering histone H3 acetylation and H3K9 dimethylation levels. In addition, KYP can directly target the promoters of *KNAT1* and *KNAT2*, and the binding of KYP depends on AS1. Furthermore, the genome-wide occupancy profile of KYP indicated that KYP is enriched in the promoter regions of coding genes, and the binding of KYP is positively correlated with that of AS1 and HDA6. Together, these results indicate that *Arabidopsis* H3K9 methyltransferases KYP/SUVH5/6 are involved in leaf development by interacting with AS1-AS2 to alter histone H3 acetylation and H3K9 dimethylation from *KNAT1* and *KNAT2* loci.

[1] Institute of Plant Biology, National Taiwan University, Taipei 10617, Taiwan. [2] RIKEN, Center for Sustainable Resource Science, Yokohama 230-0045, Japan. [3] Department of Life Science, National Taiwan University, Taipei 10617, Taiwan. [4] Guangdong Key Laboratory for New Technology Research of Vegetables, Vegetable Research Institute, Guangdong Academy of Agricultural Sciences, Guangzhou 510640, China. [5] These authors contributed equally: Fu-Yu Hung, Yun-Ru Feng. ✉email: kewu@ntu.edu.tw

The initiation of leaf primordia is established by recruitment of the cells flanking the shoot apical meristem (SAM). Meristem activity in the shoot apex is specified in part by the class I *KNOTTED-LIKE HOMOBOX* (*KNOX*) genes[1–3]. Lateral organs such as leaves are initiated on the flank of the shoot apical meristem, and down-regulation of *KNOX* gene expression is essential to facilitate this process[1,4]. Moreover, silencing of *KNOX* genes is important in developing organs since ectopic *KNOX* expression during organogenesis results in patterning defects and hyper-proliferation of cells[5–7]. In *Arabidopsis*, the members of the *KNOX* family can be divided into three classes. Class I *KNOX* genes include *BREVIPEDICELLUS/KNOTTED-LIKE FROM ARABIDOPSIS THALIANA1* (*BP/KNAT1*), *KNAT2*, *KNTA6*, and *SHOOTMERISTEMLESS* (*STM*)[8]. Class II *KNOX* genes comprise *KNAT3*, *KNAT4*, *KNAT5*, and *KNAT7*, which are broadly expressed and have been shown to function redundantly to influence lateral organ differentiation in *Arabidopsis*[9]. Class III only contains *KNATM*, which is a *KNOX* gene lacking the homeodomain[10]. In *Arabidopsis*, *KNAT1* is expressed in the vegetative meristem and stem, and is down-regulated as leaf primordia are initiated[6]. Thus, the precise balance between the differentiation and proliferation of stem cells is achieved in part through proper regulation of *KNOX* expression.

*KNOX* repression during organogenesis is mediated by the transcription complex composed of the MYB domain protein ASYMMETRIC LEAVES1 (AS1) and the AS2/LATERAL ORGAN BOUNDARIES (AS2/LOB) domain protein AS2 in *Arabidopsis*[11–16]. *KNAT1* and *KNAT2* are mis-expressed in the leaves and flowers of the *as1/as2* double mutant, suggesting that AS1 and AS2 promote leaf differentiation by repressing *KNOX*[14]. The AS1-AS2 complex (AS1/2) can recruit a chromatin-remodeling protein HISTONE REGULATORY HOMOLOG 1 (HIRA) to regulate target gene expression during organogenesis[17]. In addition, AS1/2 can also recruit POLYCOMB-REPRESSIVE COMPLEX 2 (PRC2) to repress *KNOX* genes by histone H3 lysine 27 methylation[18]. Collectively, these studies suggest that the repression activity of AS1/2 is associated with histone modifications.

Histone modifications including methylation, acetylation, phosphorylation, and ubiquitination can influence transcription, DNA repair, replication, and recombination[19,20]. Lysine methylation on the side chains of histones is regulated by histone methyltransferases (HMTs) and histone demethylases (HDMs)[19,20]. Methylation on lysine 9 and 27 of histone H3 (H3K9me and H3K27me) is associated with transcription repression, while methylation on lysine 4 and 36 of histone H3 (H3K4me and H3K36me) is associated with transcription activation[19,20]. For instance, H3K9 mono-methylation (H3K9me1) and H3K9 dimethylation (H3K9me2) mainly function in repressing transposon activities. H3K9me2 is enriched in transposons and repeated sequences[21–24]. In addition, the level of histone acetylation is controlled by histone acetyltransferases (HATs) and histone deacetylases (HDACs). HATs can add acetyl groups to lysine, which loosens the chromatin confirmation and leads to transcription activation. In contrast, removing acetyl groups from lysine by HDACs leads to condensed chromatin structure and transcription repression[19,20].

Histone lysine methyltransferases (HKMTs) have a specific conserved domain called SET (SUPPRESSOR OF VARIEGATION, ENHANCER OF ZESTE AND TRITHORAX) domain, which is mainly responsible for histone methylation activity. In *Arabidopsis*, 49 SET Domain Group (SDG) proteins have been identified, and 31 of them are known or predicted to have HKMT activity. These SDG proteins can be further classified into five classes (class I to V) based on their domain architectures or their target lysine residues[25]. Previous studies have revealed that the Class V SDG proteins including SUPPRESSOR OF VARIEGATION 3–9 HOMOLOG (SUVH) and SUPPRESSOR OF VARIEGATION 3–9 RELATED (SUVR) proteins are associated with H3K9 methylation involved in heterochromatin maintenance and DNA methylation[26–29]. All SUVH proteins contain a SET domain, a pre-SET domain, a post-SET domain, and a STE and RING-associated (SRA) domain. The SRA domain is responsible for recognizing methylated DNA[30]. KRYPTONITE (KYP, also called SUVH4), SUVH5 and SUVH6 are the best-characterized SUVH proteins in *Arabidopsis* and they function as histone H3K9 methyltransferases. KYP is required for the maintenance of CHG methylation controlled by CHROMOMETHYLASE 3 (CMT3)[31–33]. Furthermore, KYP, SUVH5, and SUVH6 act redundantly to silence transposable elements (TEs) by regulating H3K9me1 and H3K9me2 at their target loci. The *kyp/suvh5/suvh6* triple mutant displays a loss of non-CG methylation similar to the *cmt3* mutant[22,23,27,31–35]. The histone deacetylase HDA6 is also involved in transposon silencing[36]. In addition, HDA6 interacts and functions synergistically with KYP, SUVH5, and SUVH6 to co-regulate transposon silencing through histone H3K9 methylation and H3 deacetylation[37].

Although it has been established that KYP/SUVH5/6 are important regulators of *TE* silencing, their function in plant development remains elusive. In this study, we found that KYP/SUVH5/6 interacts with AS1/2 and regulates leaf development by repressing *KNAT1* and *KNAT2* expression through H3K9me2 and H3 deacetylation.

## Results

### *Arabidopsis* KYP/SUVH5/6 are involved in leaf development.
Although the involvement of *Arabidopsis* H3K9 demethylases in plant developmental processes has been reported[38–41], the function of KYP and SUVH5/6 in plant development remains elusive. Our recent study has revealed that KYP and SUVH5/6 interact with the histone deacetylase HDA6 and they function synergistically to regulate *TE* expression[37]. To further investigate the biological function of KYP/SUVH5/6, we analyzed the growth phenotypes of *hda6* and *kyp* single, *kyp/hda6* double, *kyp/suvh5/6* triple, and *hda6/kyp/suvh5/6* quadruple mutants. As reported previously[42], the *hda6* mutant had curling and serrated leaves. Compared to Col-0 wild type (WT), *hda6*, *kyp* and *kyp/suvh5/6* mutants also displayed a slight curling leaf phenotype (Fig. 1a–d, Fig. S1a). The curling leaf phenotype was enhanced in *kyp/hda6* (Fig. 1a–d, Fig. S1a) compared with *hda6* and *kyp*. Interestingly, we found a further enhanced leaf developmental defect in the *hda6/kyp/suvh5/6* quadruple mutant compared with *hda6* and *kyp/hda6*. Furthermore, the leaves of *hda6/kyp/suvh5/6* plants were also much smaller (Fig. 1a–d). Quantitative analyses indicated that nearly 80% of leaves in the *hda6/kyp/suvh5/6* quadruple mutant were developmental defective (Fig. 1e, Fig. S1a). The defective leaf phenotype of *hda6/kyp/suvh5/6* was also more severe when compared to *hda6/suvh5* and *hda6/suvh5/6* (Fig. S1b, S1c). Collectively, these results suggest that HDA6 functions synergistically with KYP/SUVH5/6 in the regulation of leaf development.

### KYP/SUVH5/6 interact with AS1/2.
Our previous study showed that *Arabidopsis* HDA6 is functionally associated with AS1/2[42]. We performed bimolecular fluorescence complementation (BiFC) assays and co-immunoprecipitation (Co-IP) assays to investigate whether KYP/SUVH5/6 can interact with AS1/2. We found that KYP, SUVH5 and SUVH6 can interact with both AS1 and AS2 in BiFC assays by using *Agrobacterium*-infiltrated tobacco leaves (Fig. 2a–c) and *Arabidopsis* protoplasts (Fig. S2a). The interaction of KYP with AS1 was further confirmed by Co-IP assays using *KYPpro::KYP:GFP/kyp* transgenic plants carrying KYP fused with GFP driven by the *KYP* native promoter. The endogenous AS1

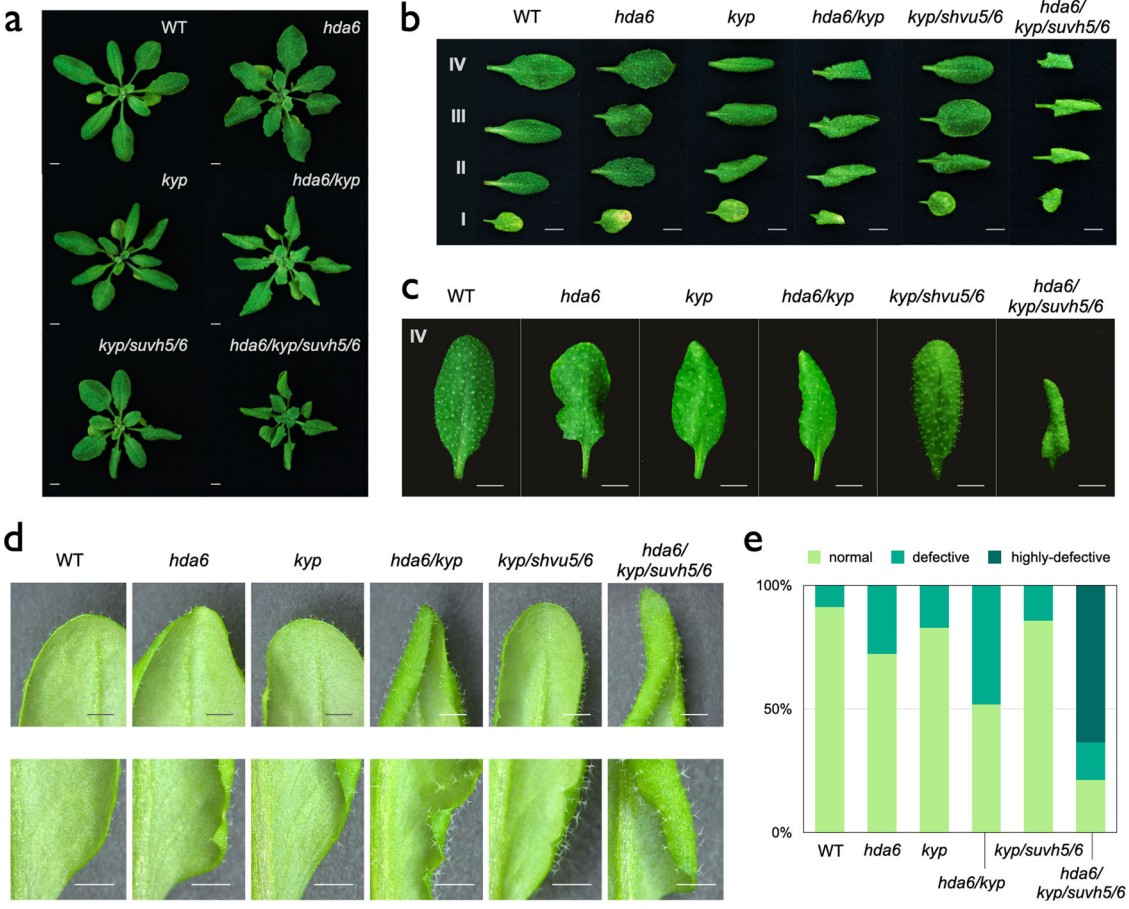

**Fig. 1 *Arabidopsis* KYP, SUVH5 and SUVH6 are involved in leaf development. a–d** Leaf development phenotypes of WT, *hda6, kyp, hda6/kyp, kyp/suvh5/6* and *hda6/kyp/suvh5/6* mutant plants. Plants were grown under long-day for 20 days. I, II, III, and IV indicate the first, second, third, and fourth pair of leaves, respectively. Bars= 5 mm (**a–c**) or 2 mm (**d**). **e** Quantitative analysis of leaf development phenotypes of WT, *hda6, kyp, hda6/kyp, kyp/suvh5/6* and *hda6/kyp/suvh5/6* plants. The fourth pair rosette leaves were classified as normal, defective and highly-defective leaves. The ratio of defective and total leaves was calculated. At least 40 leaves for each line were scored.

protein was detected in transformed plants by using an anti-AS1 antibody. As shown in Fig. 2d and S2B, AS1 interacted with KYP in Co-IP assays. In addition, the interaction between KYP with AS2-GFP was also confirmed by Co-IP assays using *KYPpro::KYP:FLAG/kyp* protoplasts (Fig. S2c).

To further confirm whether KYP can interact with AS1 and AS2 in vitro, we performed quartz crystal microbalance (QCM) assays with KYP, AS1, and AS2 recombinant proteins. The results showed that KYP interacted with AS1 and AS2 in vitro (Fig. 2e). The average Kd and standard deviation values obtained from 3 replicates of the AS1-AS2, AS1-KYP, and AS2-KYP pairs were $25.3 \pm 12.8\,\mu M$, $33.6 \pm 0.43\,\mu M$ and $18.6 \pm 13.7\,\mu M$, respectively (Fig. S2d).

Various deletion constructs of AS1 and AS2 were also generated to determine the domains responsible for their interaction with KYP using BiFC assays (Fig. 2a). Although the N-terminus of AS1 interacted strongly with KYP, the interaction between KYP and the C-terminus of AS1 was strongly decreased (Fig. 2b). Similarly, the YFP signal could be detected in the nucleus when KYP co-expressed with the N-terminus of AS2, but not with the C-terminus of AS2 (Fig. 2b). These data indicate that the N-terminus of AS1 or AS2 is responsible for the interaction.

In addition, the leaf development phenotype of the *hda6/kyp/suvh5/6* quadruple mutant displayed a defective leaf phenotype similar to *as1* and *as2* (Fig. S3a). We also generated the *as1/kyp* double, *as1/hda6/kyp* triple and *as1/hda6/kyp/suvh6*

quadruple mutant plants. Compared to WT, these mutants also displayed a defective leaf phenotype similar to *as1* (Fig. S3b), suggesting that the function of HDA6-KYP/SUVH5/6 in leaf development is at least partially dependent on AS1. Collectively, these results indicate that KYP, SUVH5, and SUVH6 are involved in leaf development by interacting with AS1 and AS2.

**KYP/SUVH5/6 repress *KNAT1/2* by altering H3K9me2 and H3Ac levels of *KNAT1/2* loci.** AS1 and AS2 are transcription repressors of the class I *KNOX* genes[17]. To investigate whether KYP and SUVH5/6 affect the expression of *KNOX* genes, we analyzed the expression of *KNAT1, KNAT2, KNAT6* and *STM* in WT, *hda6, kyp, hda6/kyp, kyp/suvh5/6* and *hda6/kyp/suvh5/6*. The expression of *KNAT1, KNAT2* and *KNAT6* was significantly increased in the mutants compared to WT (Fig. 3a). Furthermore, the highest expression levels of these class I *KNOX* genes were observed in the *hda6/kyp/suvh5/6* quadruple mutant (Fig. 3a), indicating that KYP, SUVH5/6, and HDA6 act synergistically to repress the expression of the class I *KNOX* genes.

We further investigated whether KYP/SUVH5/6 and HDA6 affected the level of H3K9me2 and H3Ac on *KNAT1* and *KNAT2* loci by chromatin immunoprecipitation followed by quantitative PCR (ChIP-qPCR). The previously identified AS1/2 binding sites (X and Y)[17] and other regions such as the promoter (P), first exon (S) and coding region (E) of *KNAT1* and *KNAT2* were selected

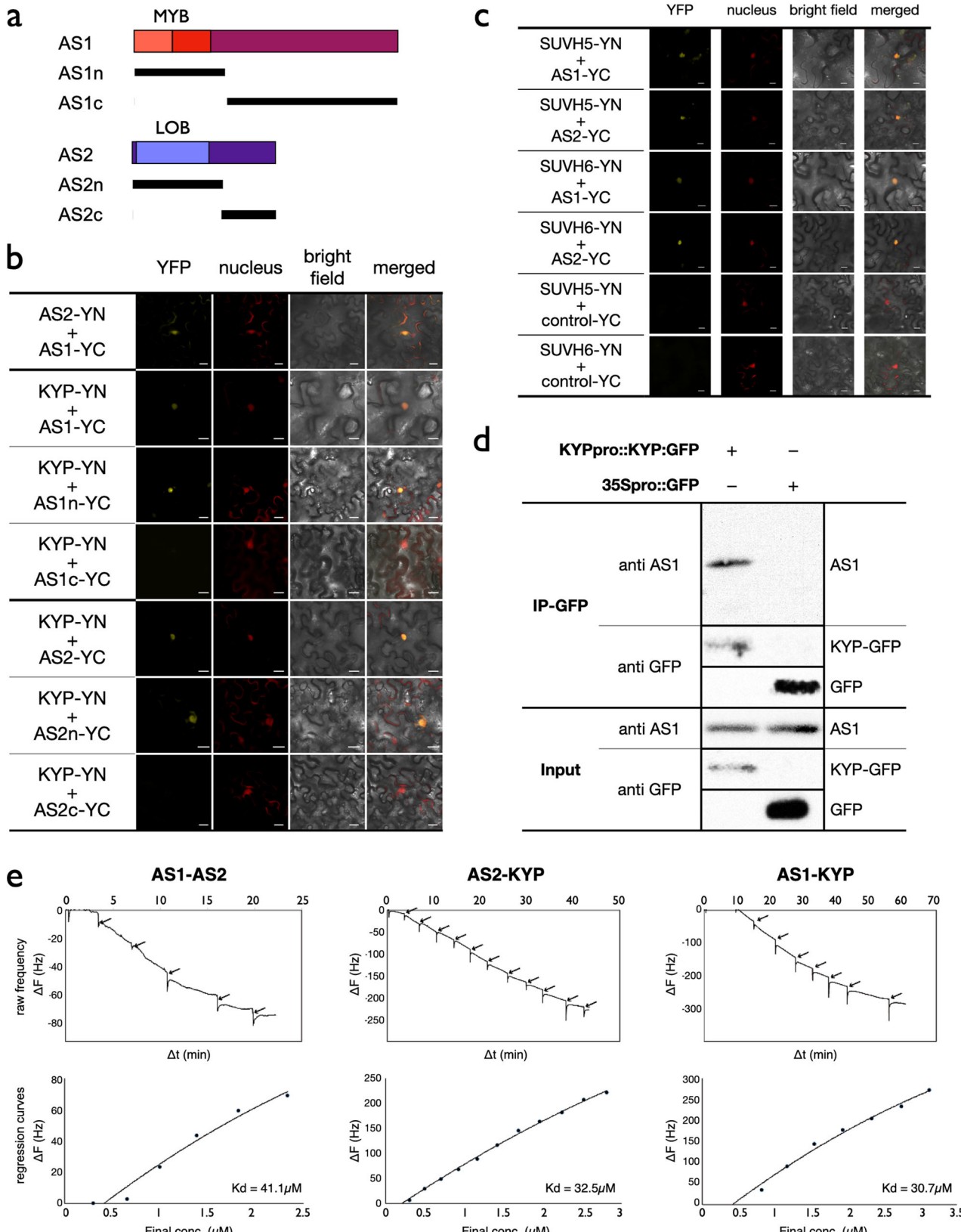

for ChIP-qPCR analysis (Fig. 3b). Compared to WT, we found that the H3K9me2 level of *KNAT1* and *KNAT2* was decreased in *kyp, hda6/kyp, kyp/suvh5/6* and *hda6/kyp/suvh5/6*, but not in *hda6* (Fig. 3c). In addition, the H3Ac level of *KNAT1* and *KNAT2* was increased in *hda6* and *hda6/kyp/suvh5/6* compared to WT (Fig. 3d). These results suggest that KYP/SUVH5/6 and

HDA6 regulate *KNAT1/2* expression through H3K9me2 and H3 deacetylation. Interestingly, we found that the H3K9me2 level of *KNAT1/2* was not decreased in *hda6* (Fig. 3c). Furthermore, the H3Ac level of *KNAT1/2* was not significantly changed in *kyp* and *kyp/suvh5/6* (Fig. 3d). The expression of *KNAT1/2* was highest in the *hda6/kyp/suvh5/6* quadruple mutant (Fig. 3a), indicating that

**Fig. 2 KYP, SUVH5 and SUVH6 interact with AS1 and AS2. a** Schematic representation of deletions in AS1 and AS2 constructs. MYB: MYB domain of AS1; LOB: LOB-domain of AS2. **b, c** BiFC assays in *N. benthamiana* leaves showing interaction of KYP (**b**), SUVH5/SUVH6 (**c**) with AS1 and AS2 in living cells. Full-length KYP, SUVH5, SUVH6, and different regions of AS1/2 were fused with the N terminus (YN) or C terminus (YC) of YFP and co-delivered into tobacco leaves by *Agrobacterium* GV3101. The nucleus was indicated by mCherry carrying a nuclear localization signal. Bars = 20 μm. **d** Co-IP assays using the *KYP* native promoter driven *KYP:GFP* (*KYPpro::KYP:GFP*) or *35Spro::GFP* in transformed *Arabidopsis*. Western blots (WB) were performed with the indicated antibodies. **e** QCM binding assays among AS1, AS2, and KYP recombinant proteins. The raw frequency curve and regression curve obtained from the AS1-AS2, AS2-KYP, and AS1-KYP pairs were presented. Arrows indicate injection points. Three replicates of AS1-AS2, AS1-KYP, and AS2-KYP pairs were performed, and data from one representative replicate for each protein pair were shown.

both decreased H3K9me2 and increased H3Ac contribute to *KNAT1/2* expression changes.

ChIP-qPCR assays were used to identify whether KYP can directly target to *KNAT1* and *KNAT2*. *KYPpro::KYP:FLAG/kyp* transgenic lines were generated, in which the *KYP* genome sequence containing the *KYP* native promoter fused with the *3xFLAG* epitope tag was transformed into the *kyp* background. Both the *KYP* transcript and KYP protein were detected in the *KYPpro::KYP:-FLAG/kyp* transgenic lines (Fig. S4). The expression of several *TEs* which are highly activated in *kyp* was analyzed by RT-qPCR. We found that these *TEs* were not activated in the *KYPpro::KYP:FLAG/kyp* transgenic plants. These results indicate that *KYPpro::KYP:-FLAG* is functional, since it complemented the *TEs* activation phenotype of the *kyp* mutant. ChIP assays were performed with an anti-FLAG antibody using *KYPpro::KYP:FLAG* transgenic seedlings and the binding of KYP to *KNAT1* and *KNAT2* was analyzed by ChIP-qPCR. We found that KYP was highly enriched in the promoter regions of *KNAT1* and *KNAT2* (Fig. 4a, b). Furthermore, the KYP-enriched promoter regions highly overlapped with the binding regions of AS1/2[17]. These results indicate that KYP regulates *KNAT1* and *KNAT2* expression by directly targeting the *KNAT1* and *KNAT2* promoters.

To further identify the functional correlation between KYP and AS1, we expressed *KYPpro::KYP:FLAG* in the *as1* mutant background (*KYPpro::KYP:FLAG/as1*). The protein levels of KYP in *KYPpro::KYP:FLAG/as1* and *KYPpro::KYP:FLAG* were similar (Fig. S4B). We found that the binding of KYP to *KNAT1* and *KNAT2* was significantly reduced in *KYPpro::KYP:FLAG/as1* (Fig. 4c), indicating that the binding of KYP to *KNAT1* and *KNAT2* is at least partially dependent on AS1. Furthermore, the H3K9me2 level of *KNAT1* and *KNAT2* was decreased but the H3Ac level was increased in the *as1/as2* mutant (Fig. S5), suggesting that AS1/2-regulated *KNAT1* and *KNAT2* expression is associated with H3K9me2 demethylation and H3 acetylation.

**Genome-wide occupancy profiles of KYP.** To investigate the genome-wide function of KYP in gene regulation, we mapped the genome-wide occupancy of KYP by chromatin immunoprecipitation followed by sequencing (ChIP-seq) using the *KYPpro::KYP:3xFLAG/kyp* transgenic line. KYP-occupied 3,924 genomic regions, including *KNAT1* and *KNAT2*. The genome browser views of the ChIP-Seq data show that KYP can target the promoters of *KNAT1* and *KNAT2* (Fig. 5a), which is consistent with our ChIP-qPCR data. Compared to the *Arabidopsis* genomic region distribution, the binding of KYP was more enriched in the 1 kb promoter regions, but less enriched in the gene bodies (Fig. 5b). In *Arabidopsis*, there are several histone H2A variants, such as H2A.X, H2A.Z, and H2A.W[43]. H2A.X and H2A.Z are associated with transcription regulation, whereas H2A.W is highly enriched in the heterochromatin region[44–46]. H2A.W can therefore be used as a heterochromatin marker[44,47]. Previous studies have indicated that KYP is responsible for silencing *TEs*, which are mainly located in heterochromatic regions[23,32,33]. We compared the binding of KYP with different H2A variants. Surprisingly, the binding patterns of KYP and H2A.W are different, indicating that in addition to the heterochromatic region, KYP

can also target to the euchromatin regions (Fig. 5c). Indeed, we found that most of the KYP-targeted genes are protein-coding genes (Fig. 5d).

We further compared the binding profiles of KYP among the protein coding genes and TE genes. In both protein coding genes and TE genes, the general binding of KYP was more enriched on the promoter but less enriched on the gene body (Fig. 5e). Furthermore, the binding of KYP is strongly enriched near the upstream regions of transcription start sites (TSS) of protein coding genes, but not in *TE* genes (Fig. 5e). These results support that the function of KYP is associated with the regulation of protein coding genes. In addition, we further compared the binding of KYP in all annotated coding genes (27420 genes) and *TEs* (31189 *TEs*) in *Arabidopsis*. We found that there was no significant difference in KYP binding in coding genes and *TEs* (Fig. 5f). However, the binding of KYP is higher in the top 10% highly targeted *TEs* compared to the top 10% highly targeted coding genes (Fig. 5f). Collectively, these results suggest that KYP function is important in the regulation of both *TEs* and protein coding genes.

**The genome-wide binding of KYP is positively correlated with AS1 and HDA6.** To further analyze the functional correlation between KYP and H3K9me2, we compared the KYP global binding pattern with the H3K9me2 ChIP-seq data of WT and *kyp* in the published dataset[48]. We found that the binding of KYP was more correlated with those genes with lower relative H3K9me2 levels in *kyp* compared to WT (Fig. 6a). In contrast, there was no correlation between the H3 level and the binding of KYP (Fig. 6a). Similar results were also obtained when we compared the relative H3K9me2 levels in *kyp/suvh5/6* and WT (Fig. S6a). We also found that among those genes showing changed H3K9me2 levels in *kyp*, the general binding pattern of KYP was substantially higher in the genes with decreased H3K9me2 than in those genes with increased H3K9me2 (Fig. S6b). These results indicate that the binding of KYP is indeed correlated with H3K9me2.

We also compared the binding of KYP with the previously published chromatin immunoprecipitation coupled with DNA microarray (ChIP-on-chip) data of AS1[49]. Plotprofile and plot heatmap analyses indicate that the center of AS1-binding regions was associated with the enrichment of KYP (Fig. 6b), supporting that KYP can be recruited by AS1 to regulate gene expression. In addition, we also compared the KYP-occupied genomic regions with our previously published HDA6 ChIP-seq data[50]. We found that the general binding of HDA6 was enriched in the center of KYP-occupied genomic regions (Fig. 6c), supporting that HDA6 interacts with KYP to synergistically regulate gene expression. Furthermore, we also compared the KYP global binding pattern with the H3Ac ChIP-seq data of WT and *hda6* from the published dataset[50]. The binding of KYP was more correlated with the genes with increased H3Ac levels in *hda6* compared to WT (Fig. 6d). Collectively, these results suggest that the binding of KYP is associated with the H3Ac changes regulated by HDA6.

In addition, several cis-elements were enriched within the KYP binding sites, including "GATGTCATGTGTATG", "RACTTYGGCTACACC" and (AG/AAG)n repeat sequences (Fig. 6e).

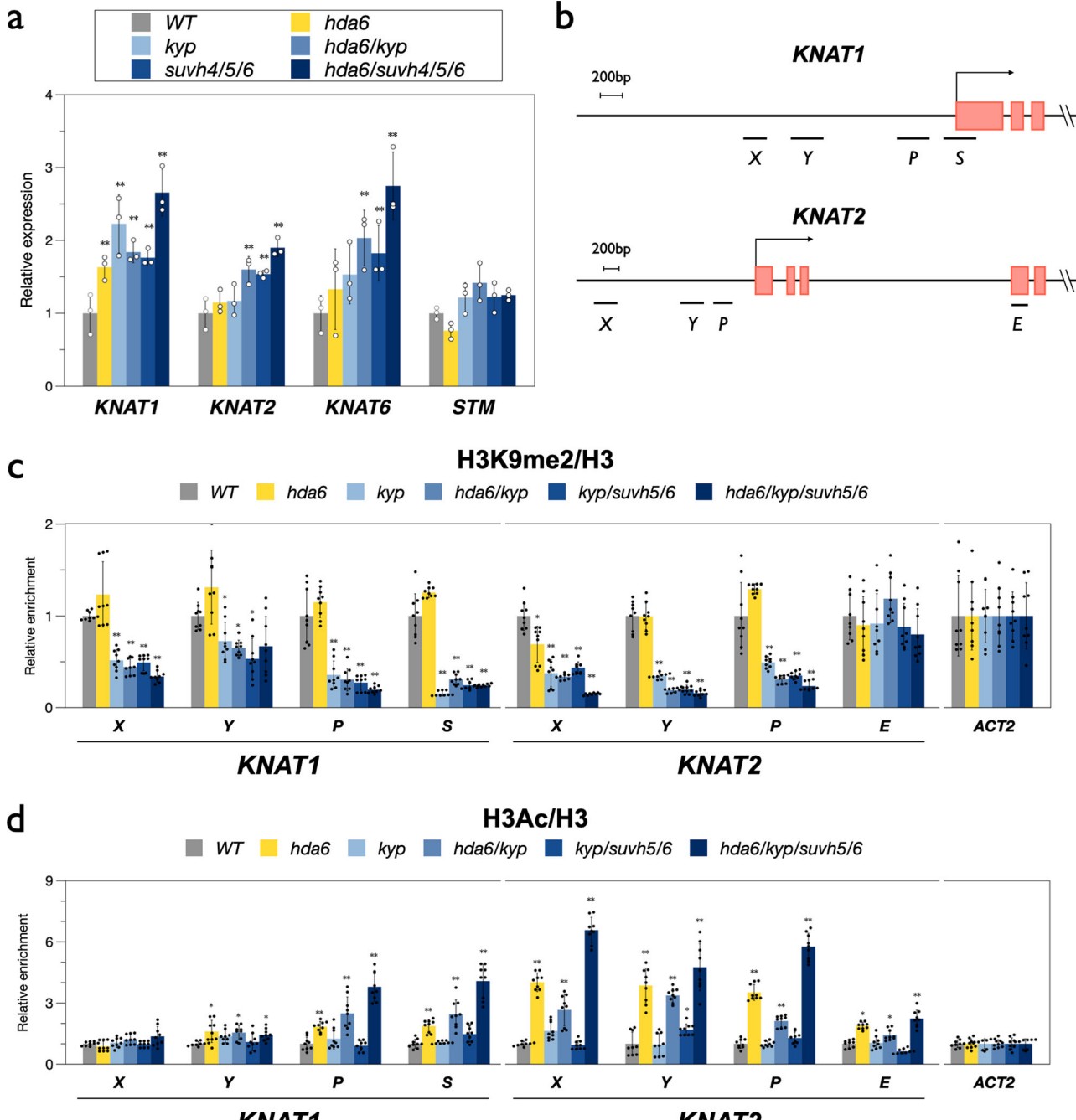

**Fig. 3 KYP, SUVH5, SUVH6 and HDA6 synergistically regulate *KNAT1* and *KNAT2* by H3K9me2 and H3 deacetylation. a** Expression of *KNOX* genes in *hda6, kyp, had6/kyp, kyp/suvh5/6*, and *hda6/suvh4/5/6* mutants. Gene expression was analyzed by RT-qPCR. RNA was extracted from 10-day-old plants grown under LD conditions. *UBQ10* was used as an internal control. Error bars indicate SD. \*\**p* < 0.01 by *t* test. At least three independent biological replicates were performed with similar results. **b** Schematic diagrams of *KNAT1* and *KNAT2* genomic sections. The promoter regions (X, Y, and P), first exon (S), and coding region (E) analyzed in ChIP-qPCR are indicated. **c, d** ChIP-qPCR analysis of H3K9me2(**c**) and H3Ac(**d**) levels on *KNAT1* and *KNAT2* in 10-day-old plants. The amounts of DNA after ChIP were quantified by qPCR. The values of relative enrichment were normalized by the value of total H3, and *ACT2* was used as internal control. Data points represent average of three technical replicates. Error bars correspond to standard deviations from three biological replicates. \**p* < 0.05, \*\**p* < 0.005 (Student's *t* test).

Interestingly, the (AG/AAG)n repeat was also found in the AS1-occupied genomic regions identified previously by ChIP-on-chip (Fig. 6e). In addition, the (AG/AAG)n repeat was also enriched in the HDA6 occupied genomic regions[50]. Together, these results support that KYP co-target on the similar genomic regions with HDA6 and AS1. The KYP-targeted genes were further analyzed according to Gene Ontology Biological Processes (GO-BP). We found that the KYP-targeted genes are involved in multiple biological processes, including abscisic acid (ABA)/stress responses and different development pathways (Fig. 6f). The involvement of KYP in ABA and stress responses has been reported[51]. Interestingly, we also found that the GO term "leaf development (GO:0048366)" was enriched in the KYP-targeted genes (Fig. 6f), supporting a role for KYP in leaf development.

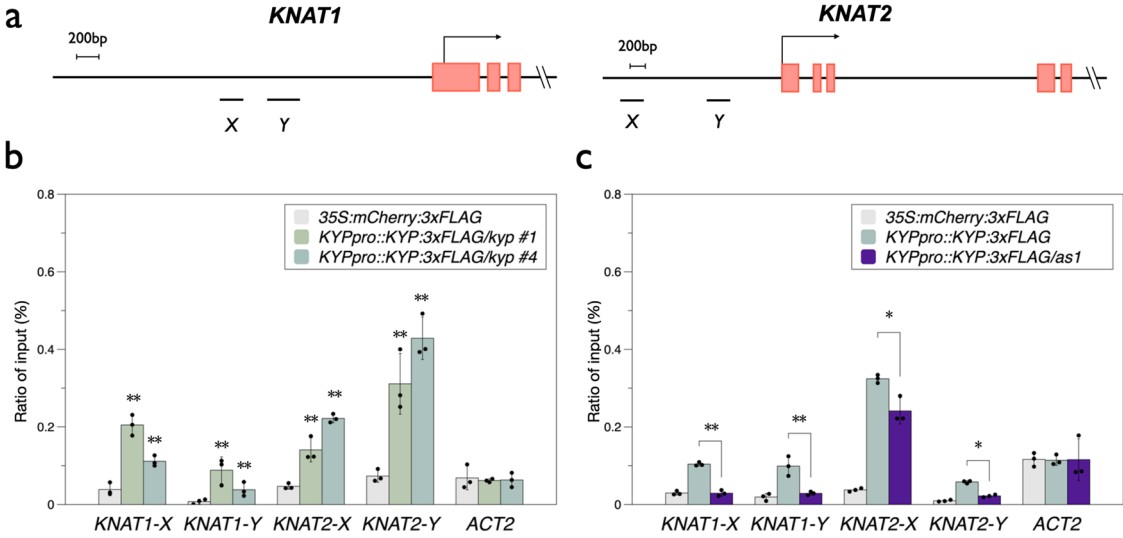

**Fig. 4 Binding of KYP to *KNAT1* and *KNAT2* in vivo. a** Schematic diagrams of *KNAT1* and *KNAT2* genomic sections. X and Y regions are the binding sites of AS1/2. **b, c** KYP binding to *KNAT1* and *KNAT2* promoters. 10 days-old seedlings grown under LD were harvested. ChIP assays were performed with the anti-FLAG antibody. The amount of immunoprecipitated DNA was quantified by qPCR. Values represent the normalized average immunoprecipitation efficiencies (%) against the total input DNA. Error bars correspond to standard deviations from three biological replicates. *$p < 0.05$, **$p < 0.005$ (Student's *t* test).

**KYP and HDA6 co-target a subset of leaf development genes.**
In addition to *KNAT1* and *KNAT2*, previous studies indicated that AS1/2 can deactivate the abaxial genes *ETTIN/AUXIN RESPONSE FACTOR3* (*ETT/ARF3*), *AUXIN RESPONSE FACTOR 4* (*ARF4*) and *YABBY5* (*YAB5*)[49,52,53]. We found that KYP and HDA6 can also target the TSS region of *ARF4* (Fig. 7a). We also identified additional genes that were targeted by KYP, such as *KNAT3, KNAT5, NUCLEOLIN 1* (*NUC1*), *GROWTH-REGULATING FACTOR 4* (*GRF4*) and *CYCLIN DEPENDENT PROTEIN KINASE 2* (*CDKC2*) (Fig. 7a). The class II *KNOX* genes *KNAT3* and *KNAT5* are involved in the development of the above-ground organs in *Arabidopsis* and *knat3/4/5* mutant plants display developmental defective leaves[9]. *Arabidopsis* NUC1 is a nucleolin protein that is involved in rRNA processing, ribosome biosynthesis, and vascular pattern formation[54]. NUC1 is also involved in leaf development and is functionally associated with AS2[55]. The transcription factor GRF4 and the cell cycle regulator CDKC2 are also involved in the regulation of leaf development[56,57]. These results indicate that HDA6-KYP/SUVH5/6 may regulate leaf development in multiple regulation pathways. In addition to leaf development, KYP target genes such as *NUC1, GRF4*, and *MLP328* are also involved in other biological processes, such as flowering, root development, stress responses and cell wall formation[58–63].

Genome browser views of the KYP ChIP-Seq data indicated that the KYP-enriched regions were highly correlated with the HDA6-enriched regions (Fig. 7a). In contrast, there are no binding peaks of KYP and HDA6 on *AT2G12520* and *ZINC RIBBON 3* (*ZR3*). In addition, we also found that the KYP and HDA6 binding sites on these target genes are substantially closed to the (AG/AAG)$_n$ repeat motif (Fig. 7a). The binding of KYP on these genes was further confirmed by ChIP-qPCR (Fig. 7b). RT-qPCR analyses indicated that the expression of these KYP-HDA6 co-targeted genes was significantly increased in the *hda6/kyp/suvh5/6* quadruple mutant compared to WT (Fig. 7c). Collectively, these results indicate that KYP and HDA6 co-targets a subset of genes involved in leaf development.

**Discussion**
In *Arabidopsis*, 31 SDG proteins predicted to have HKMT activity can be further classified into five classes (class I to class V) based on

their domain architectures or their target lysine residues[25]. There are 15 class V SDG proteins including 10 SUVH proteins and 5 SUVR proteins in *Arabidopsis*. Several class V SDG proteins have been found to be associated with H3K9 methylation involved in heterochromatin maintenance and DNA methylation[26–29]. SUVHs contain an N-terminal SRA domain and a SET domain at the C-terminus[27,64,65]. The SRA domain is required for direct binding to methylated DNA[27,65]. KYP, SUVH5, and SUVH6, the best characterized SUVH proteins in *Arabidopsis*, are H3K9me1/2 methyltransferases responsible for chromatin silencing[23,32,33]. Two other SUVH proteins, SUVH2 and SUVH9, are inactive for histone methyltransferase activity, but they can recruit RNA polymerase V to chromatin by associating with the DDR (DRD1 peptide-DMS3-RDM1) complex[66,67]. In addition, the SUVR proteins SUVR4 and SUVR5 have been found to be involved in H3K9me in vivo[68–70]. Collectively, these studies indicate that the class V SDG proteins are important in gene silencing by regulating H3K9me.

H3K9me2 is a crucial histone modification marker during embryo development in both plant and mammalian systems[26,71,72]. Recent studies have also shown that H3K9me2 is important in regulating gene expression in *Arabidopsis* development[38–41]. Although KYP and SUVH5/6 have been identified as crucial regulators of H3K9me2 in *Arabidopsis*, their function in plant development remains elusive. In the present study, we found that KYP and SUVH5/6 are functionally associated with HDA6. Furthermore, HDA6 and KYP/SUVH5/6 function synergistically to regulate the core leaf development genes, including *KNAT1* and *KNAT2*. A recent study also demonstrated that another class V SDG protein, SUVH9, is involved in embryonic development by regulating asymmetric DNA methylation[72]. Taken together, these results indicate that the Class V SDG proteins including KYP and SUVH5/6 play important roles in plant developmental processes.

*KNAT1* and *KNAT2* are class I *KNOX* homeobox genes and play important roles in meristem development and leaf morphogenesis[6–8,73]. Previous studies have demonstrated that the expression of *KNAT1* and *KNAT2* is associated with the changes in H3Ac, H3K9me2, and H3K27me3[18,42,74]. In this study, we found that KYP/SUVH5/6 and HDA6 function synergistically to regulate *KNAT1* and *KNAT2* by altering H3K9me2 and H3Ac levels. Furthermore, the expression of the *KNOX* genes was

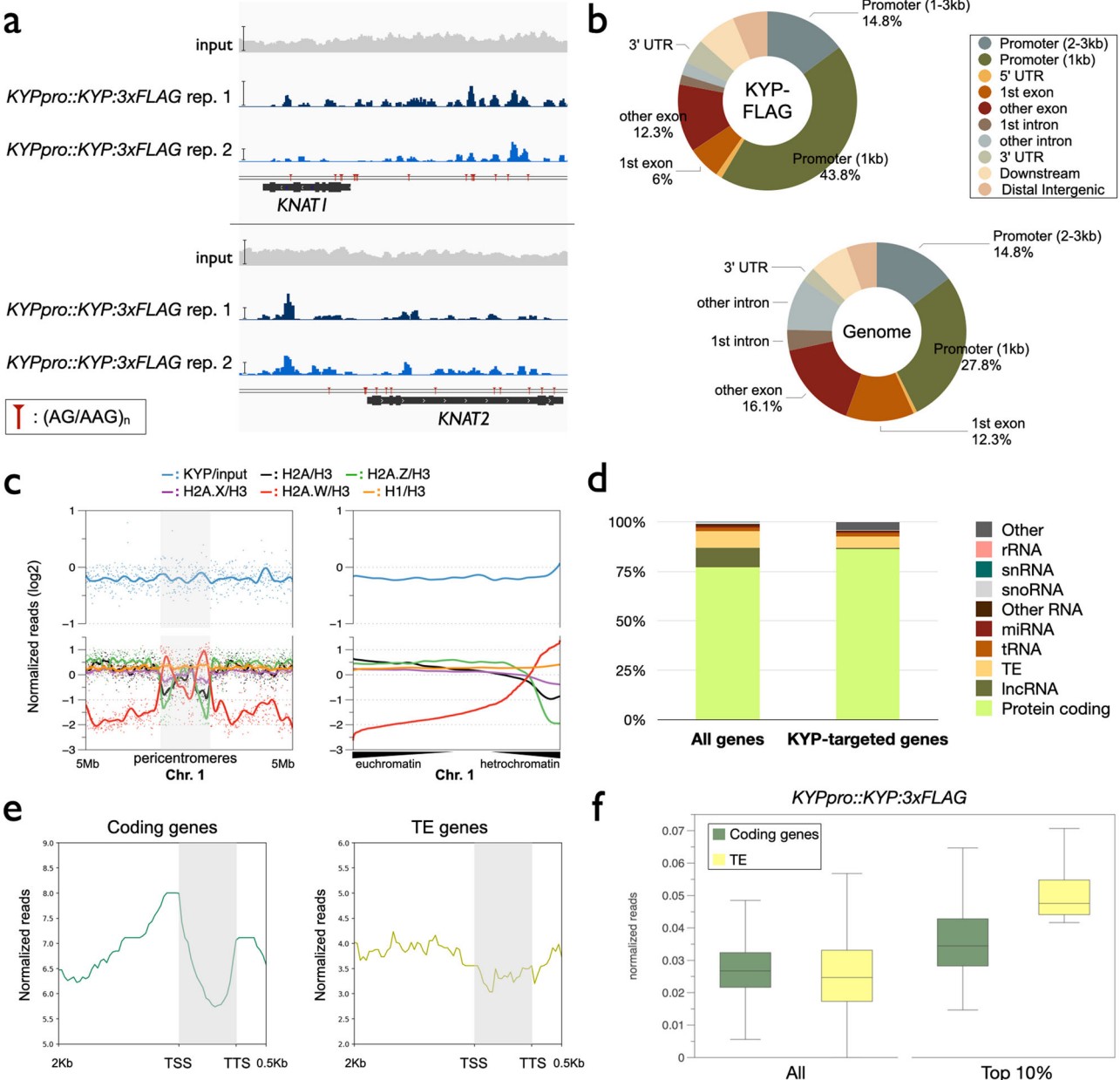

**Fig. 5 Genome-wide occupancy profile of KYP. a** Integrated Genome Viewer showing the binding of KYP on *KNAT1* and *KNAT2* genomic regions. Bars: normalized reads = 40. **b** Pie charts showing the distribution of KYP annotated genic and intergenic regions in the genome. **c** Plotprofile showing the occupancy rates of KYP, H2A, H2A.Z, H2A.X, H2A.W and H1 near the hetrochromatic region of chromosome 1 (Chr. 1). The dots indicated the log2 value of normalized reads, and the lines indicated smoothed patterns of each marker. **d** Distribution of gene types among all of the KYP-targeted genes. **e** Metagene ChIP-seq binding profiles of KYP among all coding genes and *TE* genes within the *Arabidopsis* genome. The profile is from 2 kb upstream of the TSS to 0.5 kb downstream of the TTS, and the gene lengths were scaled to the same size. **f** Boxplots showing the binding of KYP:FLAG in coding genes and *TEs*. All annotated coding genes (*n.*= 27420) and *TEs* (*n.*= 31189), or top 10% KYP:FLAG highly targeted coding genes (*n.*= 2742) and TEs (*n.*= 3119) were plotted.

increased in the *hda6/kyp/suvh5/6* quadruple mutant compared with *hda6* and *kyp/suvh5/6*. Similarly, the expression of *TEs* was also increased in the *hda6/kyp/suvh5/6* quadruple mutant compared with *hda6* or *kyp/suvh5/6*[37]. These results suggest that both H3K9me2 decreases and H3Ac increases are required for gene activation. Interestingly, it has been shown that there is an antagonistic pattern of H3K9me2 and H3Ac enrichment during embryogenesis in both plants and mammals[75,76], indicating a functional crosstalk between H3K9me2 and H3Ac in developmental processes. Our recent studies demonstrated that *Arabidopsis* HDA6 is also functionally associated with the H3K4

demethylases LDL1/2 and FLD[50,77–79]. It remains to be determined whether KYP/SUVH5/6 are also functionally associated with H3K4 demethylases.

In yeast and animal systems, HDACs are the core components of several multi-protein complexes, such as Mi2/NuRD and CoREST[80,81]. Previous studies have demonstrated that the interactions between the core protein components of Mi2/NuRD and CoREST complexes are relatively stable. However, they can dynamically interact with different transcription factors depending on environmental conditions[78,79,82,83], indicating that HDAC complexes require various transcription factors to recognize

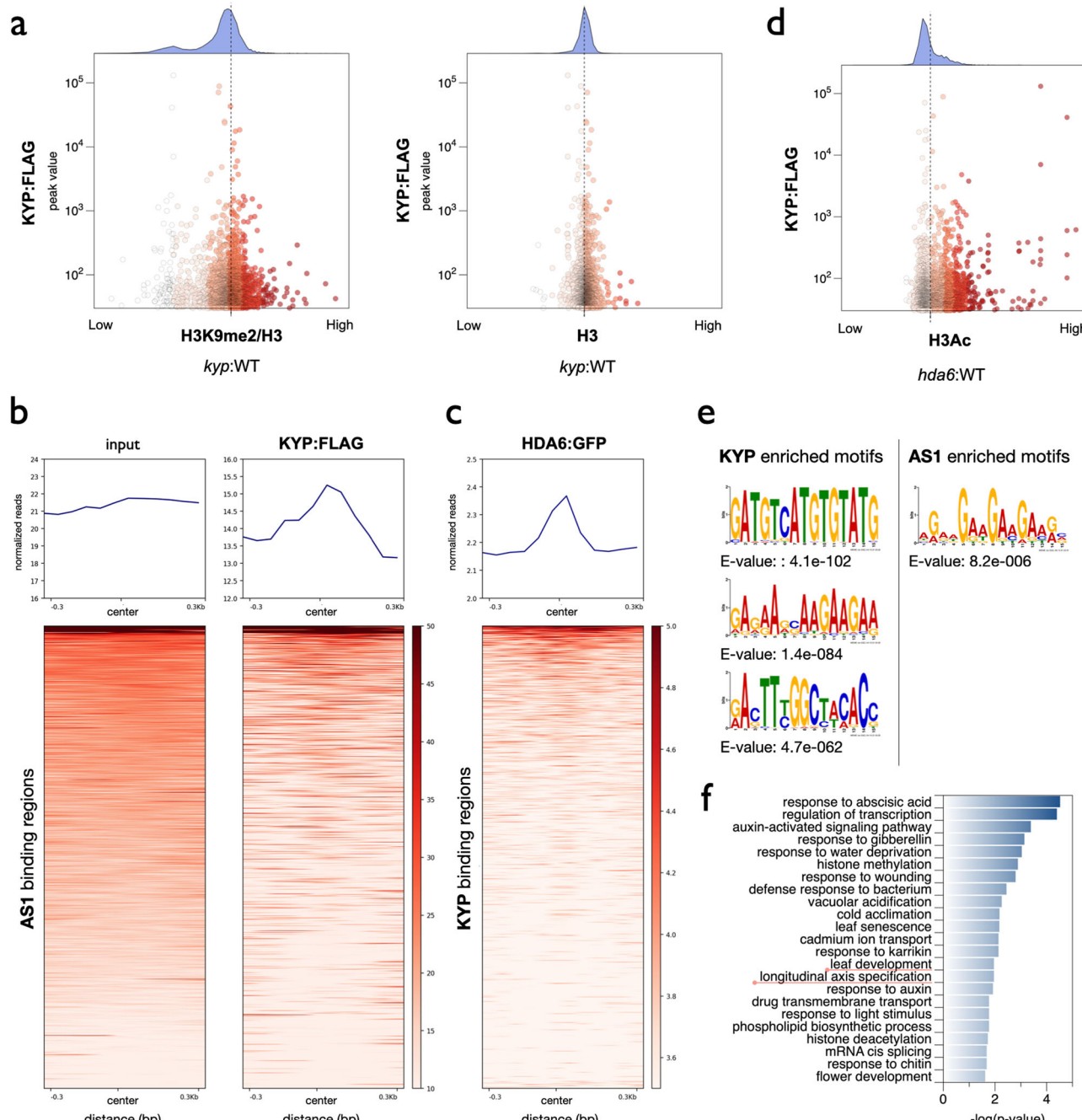

**Fig. 6 KYP-targeted genomic regions are highly correlated with AS1 and HDA6. a** X–Y scatter plots showing the relative enrichment of H3K9me2 (*kyp*: WT) and the binding level of KYP. The value of H3K9me2/H3 indicated the relative H3K9me2 level of *kyp* compared to WT and normalized to H3. The dotted line indicated the average value. **b, c** Heat map representation of the co-occupancy of KYP with AS1 (**b**) and HDA6 (**c**) in the genome. Each horizontal line represents an AS1-(**b**) or KYP-(**c**) bound region, and the signal intensity is shown for KYP (**b**) or HDA6 (**c**) binding. Columns show the genomic region surrounding each AS1-(**b**) or KYP-(**c**) peak. Signal intensity is indicated by the shade of red. **d** X–Y scatter plots showing the relative enrichment of the H3Ac level (*hda6*: WT) and the binding level of KYP. The value of H3Ac indicated the relative H3Ac level of *kyp* compared to WT. The dotted line indicated the average value. **e** DNA binding motifs significantly enriched in the KYP or AS1-binding sites. **f** GO-BP annotation of KYP -targeted genes. Annotation terms with *p*-value < 0.025 were listed.

specific genomic regions. In this study, we found that KYP and SUVH5/6 can directly interact with AS1-AS2 and regulate the expression of *KNAT1/2* by altering H3Ac and H3K9me2 levels. In addition, the binding of KYP to *KNAT1* and *KNAT2* was reduced in the absence of AS1, indicating that KYP is recruited by AS1 to the *KNAT1/2* loci.

Accumulation of H3K9me2 is highly associated with DNA methylation at CHG and CHH sites[26–29]. The triple mutant of the

non-CG DNA methylases, *drm1/drm2/cmt3* (*ddc*), is defective in leaf development with decreased hypocotyl elongation, which is associated with increased expression of the F-box domain gene *SUPPRESSOR OF drm1 drm2 cmt3* (*SDC*)[84,85]. In addition, the DNA methylation mediated by SDC is associated with periodic adjustment of circadian rhythm[85]. The involvement of HDA6-mediated histone modifications in the regulation of circadian rhythm has also been reported[78,79]. Interestingly, AS1-AS2 is also

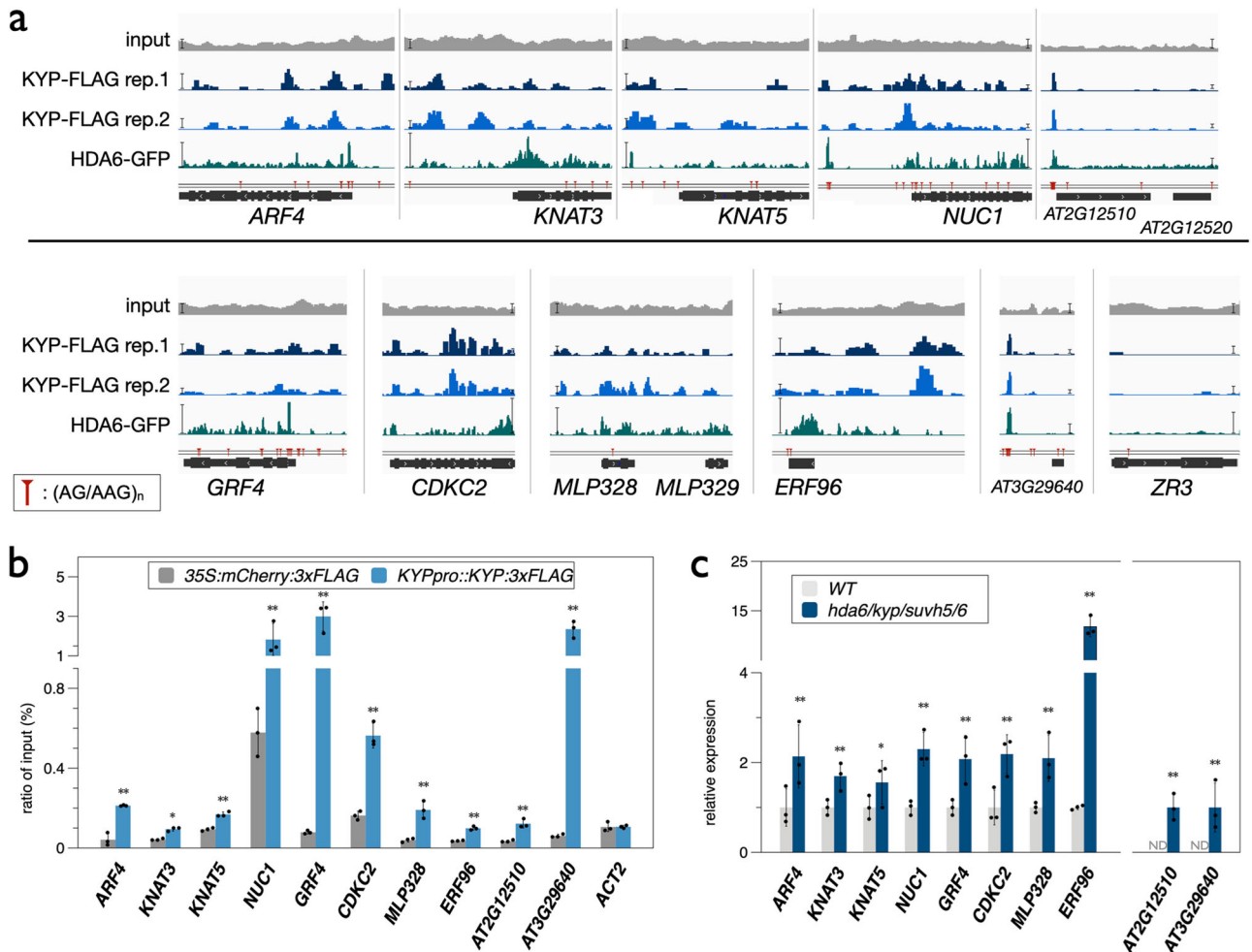

**Fig. 7 The binding and expression of KYP regulated genes. a** Integrated Genome Viewer showing the binding of KYP and HDA6 in selected genes. Bars: normalized reads = 10. Red pinpoints: (AG/AAG)$_n$ repeat motifs. **b**, **c** KYP binding (**b**) and expression (**c**) of selected genes. ChIP assays were performed with the anti-FLAG antibody. The amount of immunoprecipitated DNA was quantified by qPCR. Values represent the normalized average immunoprecipitation efficiencies (%) against the total input DNA. *UBQ10* was used as an internal control. ND: not detected. Error bars correspond to SD. *$p < 0.05$, **$p < 0.005$ (Student's *t* test). At least three independent biological replicates were performed with similar results.

required for maintaining DNA methylation on *ETT/ARF3*[49,55], suggesting that AS1-AS2 and KYP/SUVH5/6 may also function together in the regulation of DNA methylation. Recent studies indicated that AS2 is highly associated with chromocenter including ribosomal DNA repeat regions, and is involved in the regulation of cell division[86,87]. In addition, the abaxial genes *ETT/ARF3* and *ARF4* can be indirectly repressed by AS1/2 through the trans-acting siRNA (tasiRNA) called tasiR-ARFs[49,53,87]. It remains to be determined whether KYP/SUVH5/6 are also involved in these processes.

In addition to *KNAT1/2*, other leaf development genes including *KNAT3, KNAT5, NUC1, GRF4* and *CDKC2*[9,54–57] are also regulated by HDA6-KYP/SUVH5/6. GO-BP analysis indicates that KYP-targeted genes are associated with stress responses, hormone responses and different developmental processes. It has been shown that KYP is involved in regulating seed dormancy by repressing ABA signaling genes[51]. Furthermore, SUVH5 can act as a positive regulator of light-mediated seed germination[88]. Interestingly, we found that the GO-terms "response to abscisic acid" and "response to light stimulus" were also enriched in KYP-targeted genes. Taken together, these results indicate that KYP/SUH5/6 is involved in various developmental processes and pathways. By analyzing the genome-wide occupancy profile of KYP, we found that the binding of KYP was highly enriched in

promoter regions, and most of the KYP-targeted genes are protein coding genes. Furthermore, the binding of KYP is highly correlated with the binding of AS1 and HDA6. Together, these data support the notion that AS1/2 recruits the transcriptional repression complex containing HDA6 and KYP/SUVH5/6 to regulate gene expression.

In conclusion, this study provides insight into understanding how the H3K9 methyltransferases KYP and SUVH5/6 are involved in leaf development by interacting with AS1-AS2 (Fig. 8). The AS1-AS2 complex acts as a transcription repressor complex by recruiting HDA6-KYP/SUVH5/6 histone modification proteins to repress the expression of the *KNOX* genes *KNAT1* and *KNAT2* via H3K9me2 and H3 deacetylation. In addition, the HDA6-KYP/SUVH5/6 histone modification complex can also regulate gene expression involved in other developmental processes.

## Materials and methods

**Plant materials and growth conditions**. *Arabidopsis* (*Arabidopsis thaliana*) plants were germinated and grown in 22°C under long day (LD) (16 h light /8 h dark cycle) conditions. *kyp/suvh4-3*, (SALK_130630), *suvh5* (GABI_263C05), *suvh6* (SAIL_1244_F04), *hda6-6* (*axe1-5*) as well as the *kyp/suvh5/suvh6* (*kyp/suvh5/6*) triple mutant and the *hda6/kyp/suvh5/suvh6* (*hda6/kyp/suvh5/6*) quadruple mutant were reported previously[27,37,51,89]. The *hda6/kyp* double mutant was generated by

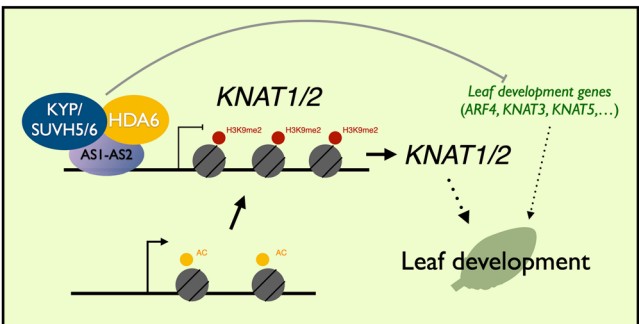

**Fig. 8 A model for the KYP/SUVH5/6 function in the regulation of *KNAT1/2* and leaf development genes.** *Arabidopsis* KYP/SUVH5/6 are involved in leaf development by repressing *KNAT1/2*. AS1-AS2 recruit HDA6-KYP/SUVH5/6 and act as transcription repressors of *KNAT1/2* by altering H3Ac/H3K9me2 levels. KYP/SUVH5/6 plays an important role in leaf development by regulating the expression of *KNAT1/2* and other leaf development genes.

crossing *kyp* (*suvh4-3*) and *hda6-6* (*axe1-5*). All mutants used in this study are in the Col-0 background.

**Plasmid construction and plant transformation**. The full-length coding sequences (CDS) of *KYP*, *SUVH5*, *SUVH6*, *AS1*, and *AS2* were reported in the previously published studies[37,42]. To generate the *KYPpro::KYP:GFP* and *KYPpro::KYP:3xFLAG* constructs, the 5 kb *KYP* genomic DNA sequence containing the 2 kb *KYP* native promoter was PCR-amplified and cloned into the *pCR8/GW/TOPO* vector (Invitrogen), then recombined into a modified *pEarleyGate302* vector containing the *3xFLAG* tag or *PMDC107* vector with the *mGFP* tag. The maltose-binding protein (MBP) fused AS1 (MBP-AS1) and AS2 (MBP-AS1) were reported previously[90]. KYP CDS was cloned into the pMAL-c5v vector to generate MBP-KYP.

*KYPpro::KYP:GFP/kyp* and *KYPpro::KYP:3xFLAG/kyp* transgenic plants were generated by transforming *KYPpro::KYP:GFP* or *KYPpro::KYP:3xFLAG* into the *kyp* mutant by the floral dip method. To express *KYPpro::KYP:3xFLAG* in the *as1* mutant background, *KYPpro::KYP:3xFLAG* plants were crossed with the *as1* mutant.

**Bimolecular fluorescence complementation and co-immunoprecipitation assays**. To generate the constructs for BiFC assays, full-length or truncated cDNA fragments of *KYP*, *SUVH5*, *SUVH6*, *AS1* and *AS2* were PCR-amplified and cloned into the *pCR8/GW/TOPO* vector (Invitrogen), and then recombined into the YN vector *pEarleyGate201-YN* and the YC vector *pEarleyGate202-YC*. Constructed vectors were transiently transformed into *Arabidopsis* protoplasts or tobacco (*Nicotiana benthamiana*) leaves. Transfected protoplasts or leaves were then examined by using a TCS SP5 confocal spectral microscope imaging system (Leica, https://www.leica.com/).

For co-immunoprecipitation assays, anti-GFP (Santa Cruz Biotechnologies, catalog no. SC-9996; 1:3000 dilution) and anti-AS1 (Luo et al., 2012; 1:3000 dilution) antibodies were used as primary antibodies for Western blot, the resulting signals were detected by using a Pierce ECL Western blotting kit (Pierce, https://www.lifetechnologies.com/).

**Quartz crystal microbalance (QCM) assays**. MBP-KYP, MBP-AS1, and MBP-AS2 recombinant proteins were expressed using *E. coli* BL21(DE3). To measure the binding ability among AS1, AS2, and KYP recombinant proteins, the quartz crystal microbalance (QCM) technique was applied. The pairwise protein-protein pairs were analyzed using an AffinixQN QCM biosensor (Initium, Tokyo, Japan). To determine Kd, one has to describe the relationship between resonance and the number of proteins on the surface undergoing adsorption by applying the Langmuir equation[91].

Prior to usage, the QCM biosensor was cleaned twice with 3 μL of piranha solution ($H_2SO_4$ and $H_2O_2$ in a 3:1 ratio) and incubated with 1% SDS for 5 min. Then, 440 μL of reaction buffer (50 mM Tris-HCl, 150 mM NaCl, and 1 mM dithiothreitol (DTT)) was applied to the dried sensor to balance and set up the magnetic stir frequency at 1000 rpm at 25 °C. For the AS1-KYP pair, 6 μL AS1 protein (2.0 mg/mL) was injected into the reaction buffer and immobilized on the Au electrode plate until saturation. Next, 4 μL KYP protein (2.0 mg/mL) was injected. The injection process was repeated until the frequency curve reaches saturation. The frequency change values were recorded as multiple binding curves using the AffinixQN v2 software (Initium, Tokyo, Japan). Data obtained from three independent repeats were processed using AQUA v2 software (Initium, Tokyo, Japan).

**Quantitative reverse transcription PCR analysis**. Total RNA was isolated using TRIZOL reagent (Invitrogen, 15596026) according to the manufacturer's instructions. Two micrograms of total RNA treated with DNAse (Promega, RQ1 #M6101) were used to synthesize cDNA (Promega, #1012891). RT-qPCR (Real-Time

quantitative PCR) was performed using iQ SYBR Green Supermix solution (Bio-Rad, #170-8880). The CFX96 Real-Time PCR Detection System (Bio-Rad Laboratories, Inc.) was used with the following cycling conditions: 95 °C for 10 min, followed by 45 cycles of 95 °C for 15 s, 60 °C for 30 s, and then fluorescent detection. This was immediately followed by a melting curve analysis (65–95 °C, incrementing 0.5 °C for 5 s, and plate reading) to confirm the absence of non-specific products. Each sample was quantified at least in triplicate, and normalized by calculating delta Cq (quantification cycle) to the expression of the internal control *Ubiquitin10* (*UBQ10*). The Cq and relative expression level are calculated by the Biorad CFX Manager 3.1 based on the MIQE guidelines. Standard deviations represent at least 3 technical and 2 biological replicates. The variance in average data is represented by SEM (standard error of the mean). The SD (standard deviation), SEM determination and P-value were calculated using Student's paired t test. The gene-specific primers used for qRT-PCR are listed in Table S1.

**Chromatin immunoprecipitation assays**. Chromatin extracts were prepared from seedlings treated with 1% formaldehyde. Chromatin was sheared to the mean length of 500 bp by sonication, proteins and DNA fragments were then immunoprecipitated using antibodies against anti-FLAG (SIGMA, catalog no. M2), H3Ac (Millipore, catalog no. 06-599), H3K9me2 (diagenode, C15410060) or total H3 (Abcam, ab1791). The DNA cross-linked to immunoprecipitated proteins were reversed, and then analyzed by real-time PCR using specific primers (Table S1). Percent input was calculated as follows: $2^{(Cq(IN)-Cq(IP))}X100$. Cq is the quantification cycle as calculated by the Biorad CFX Manager 3.1 based on the MIQE guidelines. Standard deviations represent at least 3 technical and 2 biological replicates. The variance in average data is represented by SEM (standard error of the mean). The SD (standard deviation), SEM determination and P-value were calculated using Student's paired t test.

**ChIP-seq and data analyses**. 2 ng of DNA from ChIP was pooled to ensure that there are enough starting DNA for library construction. The ChIP DNA was first tested by qPCR and then used to prepare ChIP-seq libraries. End repair, adaptor ligation, and amplification were carried out using the NEBNext® Ultra™ II DNA Library Prep kit (cat no. E7645) according to the manufacturer's protocol. The Novoseq PE150 was used for high-throughput sequencing of the ChIP-seq libraries. The raw sequence data were processed using the GAPipeline Illumina sequence data analysis pipeline. Bowtie2 was then employed to map the reads to the *Arabidopsis* genome (TAIR10)[92]. Two independent *KYPpro::KYP:3xFLAG/kyp* transgenic lines were used as biological replicates for ChIP-seq experiment. Approximately 24 and 16 million mapped reads of KYPpro::KYP:3xFLAG transgenic line #1 and #4 were used for analysis (pair-end, 150 bp). The alignments were first converted to Wiggle (WIG) files using deepTools. The data were then imported into the Integrated Genome Viewer (IGV)[93] for visualization. The distribution of the ChIP binding peaks was analyzed with ChIPseeker (supplementary data 1)[94], and a high-read random *Arabidopsis* genomic region subset (1,350,000 regions) was used to represent the ratio of the total *Arabidopsis* genomic regions. To identify DNA motifs enriched sites, 400-bp sequences encompassing each peak summit (200 bp upstream and 200 bp downstream) were extracted and searched for enriched DNA motifs using MEME-ChIP with the default parameters[95].

The KYP:FLAG ChIP-seq short read data have been submitted to the NCBI Gene Expression Omnibus (GEO) database (GSE195735).

**Statistics and reproducibility**. All graphical data represent the mean ± standard deviation of at least three biological replicates as described in figure legends. p-values calculated by paired two-tailed Student's t test were used to identify significant difference between controls and samples, as described in each figure legends.

**Reporting summary**. Further information on research design is available in the Nature Portfolio Reporting Summary linked to this article.

## Data availability
Short read data of KYP ChIP-seq have been submitted to the NCBI-Gene Expression Omnibus (GEO) database (GSE195735). The distribution of the ChIP binding peaks was provided in supplementary data 1. Un-cropped images of western-blots were provided in the Supplementary Figures. The source data to generate plots was provided in supplementary data 2.

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

## Acknowledgements

The authors thank Technology Commons, College of Life Science, National Taiwan University for the convenient use of the Apatome 2.0 microscope, TCS SP5 confocal spectral microscope imaging system and the Bio-Rad real-time PCR system. This work is supported by the National Science and Technology Council of the Republic of China (111-2311-B-002-025-MY3 and 111-2311-B-002-014 to K. W., 111-2311-B-002-016 and 111-2313-B-002-035 to Y.-S.C.), National Taiwan University 112L891801 and 112L104301 to K.W.) and Ministry of Education, Culture, Sports, and Technology of Japan (20H03284 and 20H05911 to K.S.).

## Author contributions

F.-Y.H., and K.W. designed research, Y.-R.F., F.-Y.H., K.-T.H., W.Z., C.-H.C., Y.-C.L., Y.-H.S., Y.X., and S.Y. performed research. F.-Y.H., Y.-R.F., K.-T.H., Y.-S.C. and K.W. analyzed data. F.-Y.H., Y.-R.F., K.S., Y.-S.C. and K.W. wrote the article.

## Competing interests

The authors declare no competing interests.
