## [Peer Review File · Communications Biology]

Reviewers' comments:

Reviewer #1 (Remarks to the Author):

The manuscript of Hung et al., provides novel and important insight into the regulation of KNAT1/2 expression and leaf development. The authors' conclusions are based on sound experimental evidence. The manuscript is well-written. The methods are complete. The figures are well executed. My only minor request is that the authors provide the complete blots for their Co-IP experiments as part of the supplementary material.

Reviewer #2 (Remarks to the Author):

In this manuscript, the authors confirmed that Arabidopsis KYP/SUVH5/6 are involved in leaf development by repressing KNAT1/2 through a large number of experiments such as Co-IP, ChIP-seq, BiFC, etc. Similar conclusions have been partially confirmed in previous studies, but this manuscript also provides many novel insights. Before the publication of the manuscript, the main suggestions for the improvement of the manuscript are as follows:

1. There are many grammatical errors in the manuscript that need to be carefully checked and revised.
2. From the results presented by the authors, we see that the leaf phenotype is more severe in mutants *kyp/hda6* and *hda6/kyp/suvh5/6*, whereas the phenotype is very weak in mutants *kyp* and *kyp/suvh5/6*. Did the authors investigate the phenotypes of *hda6/suvh5/6* mutants, and did their phenotypes also become more severe?
3. In Arabidopsis, KNAT1, KNAT2 and KNAT6 have similar functions, the authors examined the expression of KNAT1, KNAT2 and STM in WT, *hda6*, *kyp*, *hda6/kyp*, *kyp/suvh5/6* and *hda6/kyp/suvh5/6*, why is KNAT6 not detected?
4. Previous studies have shown that the AS1/2 complex has an inhibitory effect on KNAT1 and KNAT2 genes. In this manuscript, what is the significance of the presence of acetylation, and does it further strengthen their inhibitory effect?
5. It seems to be better if there are other in vitro experiments to further confirm the process of acetylation.

Reviewer #3 (Remarks to the Author):

The manuscript by Hung and Feng et al. investigated the roles of H3K9 methyltransferase KYP1/SUVH5/6 in regulating Arabidopsis leaf development. The authors demonstrate that KYP1/SUVH5/6 can interact with two key transcription factors, AS1 and AS2, to regulate leaf development. This study is a further extend of a previous one that shows KYP1/SUVH5/6 associate with HDA6 to silence transposable elements (TEs), showing this complex can regulate genes. The topic on how KYP/SUVH5/6 regulate leaf phenotype is of interest. However, I am not convinced by the data presented here and have several concerns that need to be clarified by the authors.

1) The authors should clarify the relationship among KYP/SUVH5/6, SDC, and AS1/2 on the phenotypes.

The authors found that *kyp* and *hda6* mutants show a phenotype of curly and serrated leaf. Then the authors link the phenotypes to AS1/2 and KNAT1/2. However, the phenotypes of these mutants are more like a DNA methylation mutant *drm1 drm2 cmt3 (ddc)*. It has been shown that the similar phenotype is due to a single gene SDC that is methylated at tandem repeats in promoter (Henderson and Jacobsen, 2008). As KYP1/SUVH5/SUVH6 is so closely linked to DNA methylation because of a reinforcing feedback loop between them and CMT3/2, *drm kyp* and *nrpd kyp* mutants also show very similar phenotypes. Thus, the authors should clarify the relationship among KYP/SUVH5/6, SDC, and AS1/2. Why did the authors even not mention SDC in the whole manuscript? Does this leaf phenotype of *hda6 kyp suvh* depend on SDC? Knocking out *sdc* in the mutants is required. A similar study

generating ddc2c3/sdc is a good example (Tian et al., 2021, Nucleic Acids Research).

2) On the mechanism. It is well known that H3K9me2 is enriched at constitutive heterochromatin, silencing TEs and other repetitive DNA. This study declares that H3K9me2 regulate a large sets of genes such as KNAT1, KNAT2 and more. This is surprising to me. I checked the ChIP-seq data of KNAT1 and KNAT2, but found no H3K9me2 peaks, except a methylated region at 3' end of KNAT1. Considering the very weak ChIP phenotype in Fig. 2C. The authors should do much more analysis to confirm H3K9me2 is indeed on this locus. Moreover, the authors should also do a parallel H3 ChIP and then normalize the H3K9me2 ChIP to a H3 ChIP. Why the authors normalize to TA3 here?

3) On the ChIP-seq data analysis. The authors performed KYP-FLAG ChIP-seq and then analyzed the genomic distribution. However, there are big problems here. The major function of KYP is to deposit H3K9me2 on heterochromatin. Thus, the authors should see TEs as the major targets. However, from figure 5C, the list of TEs used for analysis in this study seem to be not right. From TAIR, there are more than 30,000 TEs in Arabidopsis genome. The authors should reevaluate the data analysis related to this part.

4) The protein-protein interaction assay. The evidence is only based on BIFC and some CoIP. I am a little worried about the BIFC quality, as it could easily produce false positive results. Moreover, why some mCherry-NLS are located in both nucleus and cytosol? As both BIFC and CoIP are in vivo assays, some in vitro assays are required to demonstrate the interaction is direct or indirect.

5) For the immunoblots, the molecular marker is required.

6) In Fig S2B, the 'anti-AS1' should be a typo.

Reviewers' comments:

Reviewer #1 (Remarks to the Author):

The manuscript of Hung at al., provides novel and important insight into the regulation of KNAT1/2 expression and leaf development. The authors' conclusions are based on sound experimental evidence. The manuscript is well-written. The methods are complete. The figures are well executed. My only minor request is that the authors provide the complete blots for their Co-IP experiments as part of the supplementary material.

Response:

Thanks for the suggestion. We have added the complete blots in supplementary Figure S2B.

Reviewer #2 (Remarks to the Author):

In this manuscript, the authors confirmed that Arabidopsis KYP/SUVH5/6 are involved in leaf development by repressing KNAT1/2 through a large number of experiments such as Co-IP, ChiP-seq, BiFC, etc. Similar conclusions have been partially confirmed in previous studies, but this manuscript also provides many novel insights. Before the publication of the manuscript, the main suggestions for the improvement of the manuscript are as follows:

1. There are many grammatical errors in the manuscript that need to be carefully checked and revised.

Response:

Thanks for the suggestion. We have carefully checked and revised the text.

2. From the results presented by the authors, we see that the leaf phenotype is more severe in mutants *kyp/hda6* and *hda6/kyp/suvh5/6*, whereas the phenotype is very weak in mutants *kyp* and *kyp/suvh5/6*. Did the authors investigate the phenotypes of *hda6/suvh5/6* mutants, and did their phenotypes also become more severe?

Response:

Thanks for the suggestion. We have checked the phenotypes of *hda6/suvh5* and *hda6/suvh5/6*. The results have been added in Figure S1B, S1C and described on page 6, line 132.

3. In Arabidopsis, KNAT1, KNAT2 and KNAT6 have similar functions, the authors examined the expression of KNAT1, KNAT2 and STM in WT, *hda6*, *kyp*, *hda6/kyp*, *kyp/suvh5/6* and *hda6/kyp/suvh5/6*, why is KNAT6 not detected?

Response:

Thanks for the suggestion. Similar to *KNAT1/2*, we found that the expression of *KNAT6* is also increased in the *hda6/kyp/suvh5/6* quadruple mutant. The results have been added in Figure 3A, and described on page 7, line 174.

4. Previous studies have shown that the AS1/2 complex has an inhibitory effect on KNAT1 and KNAT2 genes. In this manuscript, what is the significance of the presence of acetylation, and does it further strengthen their inhibitory effect?

Response:

Thanks for the suggestion. Previous studies have shown that many histone modification enzymes cannot target to DNA by themselves, and they may need transcription factors to recognize specific genomic targets. Conversely, transcription factors recruit histone modification enzymes to change the chromatin accessibility, leading to transcription regulation. In this study, we found that KYP and SUVH5/6 can directly interact with AS1-AS2 to regulate the expression of *KNAT1/2* by altering H3Ac and H3K9me2 levels. Additionally, the binding of KYP to *KNAT1* and *KNAT2* was reduced in the absence of AS1, indicating that KYP is recruited by AS1 to *KNAT1/2* loci. We have enhanced these descriptions on the discussion section on page 14, line 364-374.

5. It seems to be better if there are other in vitro experiments to further confirm the process of acetylation.

Response:

Thanks for the suggestion. The histone acetylation/deacetylation activity of HDA6 has been reported previously (Yu et al., 2017, PMID: 28778955).

Reviewer #3 (Remarks to the Author):

The manuscript by Hung and Feng et al. investigated the roles of H3K9 methyltransferase KYP1/SUVH5/6 in regulating Arabidopsis leaf development. The authors demonstrate that KYP1/SUVH5/6 can interact with two key transcription factors, AS1 and AS2, to regulate leaf development. This study is a further extend of a previous one that shows KYP1/SUVH5/6 associate with HDA6 to silence transposable elements (TEs), showing this complex can regulate genes. The topic on how KYP/SUVH5/6 regulate leaf phenotype is of interest. However, I am not convinced by the data presented here and have several concerns that need to be clarified by the authors.

1) The authors should clarify the relationship among KYP/SUVH5/6, SDC, and AS1/2 on the phenotypes.

The authors found that *kyp* and *hda6* mutants show a phenotype of curly and serrated leaf. Then the authors link the phenotypes to AS1/2 and *KNAT1/2*. However, the phenotypes of these mutants are more like a DNA methylation mutant *drm1 drm2 cmt3 (ddc)*. It has been shown that the similar phenotype is due to a single gene SDC that is methylated at tandem repeats in promoter (Henderson and Jacobsen, 2008). As KYP1/SUVH5/SUVH6 is so closely linked to DNA methylation because of a reinforcing feedback loop between them and CMT3/2, *drm kyp* and *nrpd kyp* mutants also show very similar phenotypes. Thus, the authors should clarify the relationship among KYP/SUVH5/6, SDC, and AS1/2. Why did the authors even not mention SDC in the whole manuscript? Does this leaf phenotype of *hda6 kyp suvh* depend on SDC? Knocking out *sdc* in the mutants is required. A similar study generating *ddc2c3/sdc* is a good example (Tian et al., 2021, Nucleic Acids Research).

Response:

Thanks for the suggestions. Interestingly, we noticed a recent published RNA-seq result of the *ddc* triple mutant (Liu et al., 2020, PMID: 32493925) showing that the expression of AS1 and AS2 is decreased in *ddc*. In addition, we found that the expression and H3Ac of SDC are increased in our *hda6* RNA-seq/ChIP-seq data (Hung et al., 2020, PMID: 33575615). These data suggest a possible functional correlation in leaf development between the DDC-SDC module and the HDA6-KYP/SUVHs-AS1/2 module. The detailed functional correlation between these genes requires further studies to clarify. We have sited the suggested

references and discussed their functional correlation in the discussion section on page 14, line 375-383.

2) On the mechanism. It is well known that H3K9me2 is enriched at constitutive heterochromatin, silencing TEs and other repetitive DNA. This study declares that H3K9me2 regulate a large sets of genes such as *KNAT1*, *KNAT2* and more. This is surprising to me. I checked the ChIP-seq data of *KNAT1* and *KNAT2*, but found no H3K9me2 peaks, except a methylated region at 3' end of *KNAT1*. Considering the very weak ChIP phenotype in Fig. 2C. The authors should do much more analysis to confirm H3K9me2 is indeed on this locus. Moreover, the authors should also do a parallel H3 ChIP and then normalize the H3K9me2 ChIP to a H3 ChIP. Why the authors normalize to TA3 here?

Response:

Thanks for the suggestions. There are several other examples showing that the change of H3K9me2 detected by ChIP-qPCR is associated with coding gene expression in published papers, such as *ESP4*, *MSP2* (Kim et al., 2014, PMID: 25009302), *SHOC1*, *ZIP4* (Cheng et al., 2022, PMID: 35638341), *GOLS2*, *RD20* (Wang et al., 2022, PMID: 34197643), *WRKY25*, *FLC* and *CO* (Dutta et al., 2017, PMID: 28650521). Similar to our results, there are also no significant peaks within the H3K9me2 ChIP-seq results with these genes. Currently, how H3K9me2 affects the expression of the coding genes remains largely unknown. It could be associated with cross-talk with other histone modification markers, and gene expression may be affected by the background-level changes of H3K9me2. Additionally, we have also modified our ChIP-qPCR results by normalizing the H3K9me2 ChIP to H3 ChIP. The results have been presented in figure 3C, 3D.

3) On the ChIP-seq data analysis. The authors performed KYP-FLAG ChIP-seq and then analyzed the genomic distribution. However, there are big problems here. The major function of KYP is to deposit H3K9me2 on heterochromatin. Thus, the authors should see TEs as the major targets. However, from figure 5C, the list of TEs used for analysis in this study seem to be not right. From TAIR, there are more than 30,000 TEs in Arabidopsis genome. The authors should reevaluate the data analysis related to this part.

Response:

Thanks for the suggestions. Previously, we also expected a similar results of KYP-FLAG binding patterns as you described. However, our data show that in addition to the heterochromatic region, KYP can also target the euchromatin region. To confirm the results, we compared the binding of KYP with the heterochromatic marker H2A.W, and we found that the binding pattern of KYP was widely spread on both euchromatin and heterochromatin regions (Fig. 5C). Additionally, we further compared the binding of KYP in all annotated coding genes (n. = 27420) and TEs (n. = 31189) in *Arabidopsis*. We found that there was no significant difference in KYP binding in the coding genes and TEs (Fig. 5F). However, the binding of KYP is higher in the top 10% highly targeted TEs compared to the top 10% highly targeted coding genes (Fig. 5F). Collectively, these results suggest that KYP function is important in the regulation of both TEs and protein coding genes. These results have been presented in Figure 5C, 5F and described on page 9, line 226 and page 10, line 242.

4) The protein-protein interaction assay. The evidence is only based on BIFC and some CoIP. I am a little worried about the BIFC quality, as it could easily produce false positive results. Moreover, why some mCherry-NLS are located in both nucleus and cytosol? As both BIFC and CoIP are in vivo assays, some in vitro assays are required to demonstrate the interaction is direct or indirect.

Response:

Thanks for the suggestions. We observed both nucleus and cytosol localization of mCherry-NLS in some cases, but still good enough to recognize the nucleus in most of the cases. Additionally, we have used the truncated protein as the negative control in the BiFC assays and confirmed that the YFP signal cannot be detected in the same experiment setting. This should eliminate the false-positive concerns. To confirm whether KYP can interact with AS1 and AS2 *in vitro*, we performed the quartz crystal microbalance (QCM) assays. The results showed that KYP interacted with AS1 and AS2 *in vitro*. The results have been presented in Figure 2E, and described on page 7, line 149.

5) For the immunoblots, the molecular marker is required.

Response:

Thanks for the suggestion. We have added the western blot with the molecular marker in supplementary figure S2B.

6) In Fig S2B, the 'anti-AS1 'should be a typo.

Response:

Thanks. It has been corrected in the revised manuscript.

REVIEWERS' COMMENTS:

Reviewer #3 (Remarks to the Author):

The authors have improved the manuscript according to suggestions. However, I still have several comments that need to be considered.

1. Why there are two set of K_d values of AS1-AS2, AS2-KYP, AS1-KYP in Fig. 2E and Fig. S2D measured by QCM assays, and the values are different?
2. There are no scale bars in the microscopy images of Fig. 1B,C and S2A.
3. Figure "S7" should be "S6".

REVIEWERS' COMMENTS:

Reviewer #3 (Remarks to the Author):

The authors have improved the manuscript according to suggestions. However, I still have several comments that need to be considered.

1. Why there are two set of Kd values of AS1-AS2, AS2-KYP, AS1-KYP in Fig. 2E and Fig. S2D measured by QCM assays, and the values are different?

Response:

Thanks for the suggestion. The Kd value in Fig. 2E is only for one replicate of the QCM assay, and the average Kd and standard deviation values obtained from 3 replicates of the AS1-AS2, AS1-KYP, and AS2-KYP pairs are presented in Fig. S2D. We have changed the description in the main text and figure legends to make it more clear.

2. There are no scale bars in the microscopy images of Fig. 1B,C and S2A.

Response:

Thanks for the suggestion. The scale bars have been added in Fig. 1B, 1C and S2A.

3. Figure "S7" should be "S6".

Response:

Thanks. It has been corrected in the revised manuscript.